# A Comparison of Bispectral Index and Entropy During Sevoflurane Anesthesia Induction in Children with and without Diplegic Cerebral Palsy

**DOI:** 10.3390/e21050498

**Published:** 2019-05-15

**Authors:** Young Sung Kim, Young Ju Won, Hyerim Jeong, Byung Gun Lim, Myoung Hoon Kong, Il Ok Lee

**Affiliations:** Department of Anesthesiology and Pain Medicine, Korea University Guro Hospital, Seoul 08308, Korea

**Keywords:** cerebral palsy, consciousness monitors, electroencephalography, sevoflurane

## Abstract

Background: This study compared the correlation of bispectral index (BIS) or entropy with different sevoflurane concentrations between children with and without cerebral palsy (CP) during induction. Methods: For eighty-two children (40 CP and 42 non-CP children), anesthesia was induced with sevoflurane. BIS and entropy (response entropy and state entropy (RE and SE)) were recorded before and after the induction of anesthesia at end-tidal sevoflurane concentrations of 1–3 vol%. The sedation status was assessed using an Observer’s Assessment of Alertness/Sedation scale. The ability to predict awareness was estimated using the area under the receiver-operator characteristic curve (AUC) analysis. Results: RE, SE and BIS values decreased continuously over the observed concentration range of sevoflurane in both groups. The SE values while awake and the RE, SE, BIS values at 3 vol% sevoflurane were lower in children with CP than in those without CP. The AUC of the BIS was significantly better than RE or SE in children without CP. The AUC of the BIS was not significantly higher than that of the RE or SE in children with CP. Conclusion: BIS seems better correlated than entropy with the clinical state of loss of response in children without CP, but not in those with CP.

## 1. Introduction

Monitoring the depth of anesthesia (DOA) has proven difficult. The two monitors which are commercially available, the bispectral index (BIS) monitor, providing the BIS values, and the entropy, providing the response entropy (RE) and state entropy (SE) values [1], have disturbingly large inter-individual variation and lack of linearity in dose–response [2]. A recent study suggest that the prediction probabilities (*P*_K_) for BIS, RE and SE were similar in pediatric patients [3]. A previous study showed that children with cerebral palsy require less propofol to reach a BIS of 35–45 than healthy children [4]. Although, Choudhry and Brenn [5] showed the reliability of BIS in children with cerebral palsy, few studies have analyzed the effects of entropy in cerebral palsy. This study aimed to compare the validity between the BIS and entropy in cerebral palsy.

We hypothesized that there would be no difference in BIS or entropy values between two groups of children with or without cerebral palsy. Based on the outcome of a previous study, the correlation of BIS or entropy values with different sevoflurane concentrations during induction was set as the primary endpoint. Other observations (incidence of excitement during sevoflurane induction, end-tidal sevoflurane concentration at loss of response, and different values of BIS or entropy at loss of response) were set as secondary endpoints. The predictive power in discriminating the loss of response state from the awake state was the tertiary endpoint. To test this hypothesis, we compared the correlation of BIS or entropy values with different sevoflurane concentrations during induction between children without cerebral palsy and those with diplegic cerebral palsy.

## 2. Materials and Methods

The study was approved by the Institutional Review Board of Korea University Guro Hospital (GR0832-002). The study objective and methods were clearly explained to all parents/guardians and written informed consent was obtained. We prospectively studied children who had normal intelligence with or without diplegic cerebral palsy with American Society of Anesthesiologists physical status I or II and scheduled for elective orthopedic surgery. The exclusion criteria were a history of anti-epileptic medication; mental retardation; cardiac, pulmonary, or renal disease; and contraindication for an inhalational induction. Due to the low incidence of cerebral palsy, we included all cerebral palsy (CP) patients who visited during the experimental period. Each non-CP patient who visited next to a CP patient was included for matching the 1:1 number of patients.

All patients were sedated 30 min before anaesthesia induction with midazolam 0.5 mg/kg intramuscularly, and underwent electrocardiography, pulse oximetry, and non-invasive blood pressure monitoring. BIS values were recorded on an aspect medical system monitor (model A2000; Natick, MA, USA; smoothing time, 15 s) using commercially available BIS-XP sensor strips (Aspect Medical Systems, Newton, MA, USA). An entropy sensor (M-entropy plug-in Module S/5^TM^; Datex-Ohmeda Division, Instrumentarium Corp., Helsinki, Finland) was used for collecting entropy values. After sterilizing the forehead skin using alcohol swabs, the sensors were placed in close proximity. The temporal electrodes for BIS and entropy measurements were placed to the right and left, respectively. Two sensors were placed in the immediate vicinity of each other: the entropy sensor lower on the forehead and the BIS sensor higher. The BIS uses a 100-point index score, wherein values close to 100 indicate the awake state, and values close to 0 indicate very deep general anaesthesia. Entropy values were in the range of 0–91 for state entropy, or 0–100 for response entropy.

Anesthesia was induced using sevoflurane inhalation alone via a tight-fitting facemask and head strap held by a single experienced pediatric anesthesiologist. We confirmed that air leakage from the margin of the facemask was minimal when positive or negative airway pressure was applied. The facemask was connected to a semi-closed circle system with a fresh gas flow of 6 L/min of 100% oxygen. The end-tidal sevoflurane concentrations and carbon dioxide concentrations were continuously monitored using the S/5^TM^ compact monitor. Baseline measurements were made after resting 5 min in the supine position; thereafter, sevoflurane was administered continuously and its concentration was increased once every 5 min in increments of 1% to achieve a steady state of end-tidal sevoflurane concentrations of 1%, 2%, and 3%. We maintained partial pressure of CO_2_ (EtCO_2_) of 35–40 mmHg. When respiratory depression or CO_2_ retention had presented, mask ventilation was assisted to the patients. Loss of consciousness (LOC) was determined as the loss of response to loud verbal commands (LOR) by name and shaking on the shoulder every 10 s. The Observer’s Assessment of Alertness/Sedation rating (OAA/S) scale was used: score 5 = awake and responds readily to the name spoken in a normal tone; 4 = lethargic response to the name in a normal tone; 3 = responds only after the name is called loudly and repeatedly; 2 = responds only after the name is called loudly and after mild shaking; and 1 = does not respond when the name is called and mild shaking. All assessments of sedation level were performed by one investigator (Lee I.O.) to minimize inter-observer variability. At the end of the study period, measurements were discontinued, and anesthesia was deepened using an appropriate dose of propofol and neuromuscular blocking agents; thereafter, the patients were intubated for surgery. All values from the anesthetic monitoring (including BIS and entropy scores) were recorded every 5 s to a personal computer.

Preliminary data analysis suggested that excitement behaviors or loss of responsiveness would occur between 1% and 2% end-tidal sevoflurane concentrations. Additional observations were added for 1.5% to estimate more clearly the excitatory effect of sevoflurane in patients showing excitement during the study period. These children were excluded from the following DOA assessments. Other children without excitement behavior were also included in the study.

A power analysis suggested that a minimum sample size of 40 patients for each group would be required with a significance level of 5% to achieve a power of 80%. It was calculated from SE values at 3% sevoflurane in preliminary data (effect size d = 0.636 calculated from 38.6 ± 14.3 of SE in CP versus 46.3 ± 9.4 of SE in non-CP patients). To allow for an exclusion rate, the study population was prospectively set at 90 patients.

Statistical analysis was performed using Sigma Stat for window v3.01 (2004 Systat software, Inc, GmbH, Germany). The analyzed data were tested for normality using the Kolmogorov–Smirnov test. A parametric or non-parametric analysis was performed depending upon the results of the Kolmogorov–Smirnov analysis. Data expressed as mean ± SD were compared using independent *t*-tests or Mann–Whitney U tests. Additionally, data expressed as the number of patients were compared using chi-square analysis or Fisher’s exact test, where appropriate. Statistical significance was defined as *p* < 0.05; all data are represented as mean (standard deviation) or median (interquartile range).

Receiver-operator characteristic (ROC) analysis and comparison of ROC curves were performed using Sigma Plot for Windows v11.0 (2008 Systat Software, Inc). ROC curve analysis is a technique used to assess the overall ability of a “test” to differentiate between normal and abnormal populations of results. The “tests” investigated in this study were BIS, RE and SE. Data were sampled from representing responsiveness present, that is, awake and up to LOR during sevoflurane induction.

## 3. Results

Forty children with cerebral palsy (Group CP) and 42 without cerebral palsy (Group C) were finally enrolled. Eight patients (five for Group CP and three for Group C) were excluded from this study because of the lack of informed consent. The most type of CP was spastic (31 patients; 77.5%). Seven patients (17.5%) had dyskinetic CP and the other two patients (5%) had ataxic CP. In the Group C, there were two patients with achondroplasia. The other 40 patients were of American Society of Anesthesiologists’ physical status classification 1 (ASA 1 refers to a normal healthy patient) and had no medical history. Their demographic data showed no significant differences in age, sex, weight and height (Table 1).

The baseline value of SE was significantly lower in Group CP than in Group C. During sevoflurane induction, excitement was noted in 26 children (62%) in Group C and 24 (60%) in Group CP. The end-tidal sevoflurane concentration at the beginning of excitement was significantly higher in Group C than in Group CP. However, the end-tidal sevoflurane concentration at LOR showed no significant intergroup difference. The RE, SE, and BIS values at 3% end-tidal sevoflurane concentration were significantly lower in Group CP than in Group C (Table 2).

The OAA/S scale abruptly decreased within ten minutes in the both groups. The OAA/S values at 3, 4, 5, 6 min in Group CP were significantly lower than those in Group C (P = 0.011, 0.002, 0.021 and 0.039, respectively) (Figure 1).

Plotting the end-tidal sevoflurane concentration against entropy and BIS values revealed a hysteresis in both groups (Figure 2A–C).

The areas under the ROC curve (AUCs) of both entropy values were significantly higher in Group CP than in Group C (*p* = 0.04). The intergroup difference between the AUC of BIS values was not statistically significant. The AUC of the BIS was significantly higher than that of response entropy (*p* = 0.007) or state entropy (*p* = 0.001) in Group C (Table 3; Figure 3A,B).

The cut-off values to discriminate whether LOR was presented or not were 91.5 for RE, 84.5 for SE, and 75.5 for BIS in Group C; the corresponding values in Group CP were 75.5, 73, and 65.5, respectively (Table 3; Figure 4A,B).

## 4. Discussion

In this study, we compared the BIS and entropy in children with cerebral palsy. The baseline state entropy was significantly lower in children with cerebral palsy than in those without, before anesthetic induction. Cerebral palsy develops from an injury to the developing brain, and has features of motor impairments stemming from non-progressive brain malfunction early in life [6,7]. Therefore, EEG and structural cranial abnormalities is not rare in epileptic CP [8]. Moreover, even children with seizure-free cerebral palsy show electroencephalographic (EEG) abnormalities including asynchronous slow wave, epileptiform activity, and hypsarrhythmia [9]. This reflects heterogeneity of the neural generator in the underlying disease process. In the awake state, state entropy was significantly lower in children with cerebral palsy than in those without, while response entropy was not. This can be explained by previous findings [5] showing that non-sedated children with cerebral palsy had high muscle tone with a strong electromyographic (EMG) signal. State entropy reflects only the hypnotic level from EEG, but response entropy includes EEG and forehead EMG components. Even if the children with cerebral palsy received premedication in this study, response entropy could reflect both EEG and EMG components throughout the study period.

Choudhry and Brenn [5] have reported a similar pattern of BIS change in children with quadriplegic cerebral palsy. However, they reported lower absolute BIS values in children with cerebral palsy than in normal children while awake and at different sevoflurane concentrations. Costa et al. [10] also observed lower baseline BIS values and a slower return to the conscious state in children with cerebral palsy than in the control group during recovery of anaesthesia. In our study, BIS values in the awake state were similar between children with and without cerebral palsy. This discordance in results may be attributed to a difference in the type of cerebral palsy, which was diplegic and less severe in our study.

The BIS and entropy values at 3% end-tidal sevoflurane concentration was significantly lower in children with cerebral palsy than in those without, as seen in a previous study [5]. Moreover, the excitement events occurred earlier in children with cerebral palsy, despite the lower end-tidal sevoflurane concentration. These findings suggest children with cerebral palsy are susceptible to anesthesia [4,11]. Excitement during sevoflurane induction has been previously reported [12,13], while its cause remains unknown, it may be similar to the excitement phase with diethyl ether induction. The incidence of excitement was 35–38%, and excitement at induction occurred more frequently in children when sevoflurane was used with oxygen alone rather than when combined with nitrous oxide and oxygen [12]. In our study, excitement occurred more frequently (60% in children with cerebral palsy and 62% in those without) than previously reported. We performed very slow induction (approximately 20 min to reach 3% end-tidal sevoflurane concentration) with oxygen without nitrous oxide, and think this may have elicited more excitement.

In our study, the AUC of the BIS at LOR was higher than that of response entropy or state entropy in children without cerebral palsy. This finding was also different from that of other prior studies on adults [14,15], which showed that at the point of LOC, the entropy and BIS systems did not differ significantly, suggesting that the BIS and entropy monitoring capacities on the level of sedation are similar [14]. Higher entropy values for deep sedation and surgical anesthesia were reported in younger children (3–6 years) than in older children [16]. In a recent study on adults, state entropy showed more false classification than did the BIS in the transition state from consciousness to unconsciousness, especially in sevoflurane anesthesia rather than propofol anesthesia [17]. This is because EEG signal analysis shows that the high-frequency signal and eye blink cause errors in the degree of consciousness. State entropy may be influenced more by the delta band than is the BIS, and delta band power was relatively lower in cases misclassified using raw EEG signals.

Unlike in the control group, the children with cerebral palsy in our study showed no difference in the AUCs of the BIS and entropy. Kułak and Sobaniec [18] showed a significant decrease of power alpha at occipital derivations, significant increase of theta power and delta bands, and significant decrease of interhemispheric coherence values in children with spastic diplegia. As mentioned above [17], the delta band has a greater influence on state entropy than on the BIS. Therefore, we assumed that entropy values disagreed with clinical assessment in the normal population, which has a relatively lower delta band; however, in children with diplegic cerebral palsy, this could be offset by the increased delta band.

Our study has several limitations. First, CP types in the study were not homogeneous. Although the most type was spastic, the other types, including dyskinetic and ataxic CP, may influence the results and make the interpretation difficult. However, since all the children had not taken anti-epileptic medications, the effect might not be significant. Second, the algorithms for BIS and spectral entropy calculation are sophisticated and the precise BIS algorithm is not available in the public domain [19]. Therefore, we cannot precisely explain the theoretic mechanisms underlying our results. Last, during sevoflurane induction, only the LOR was recorded as the patients experienced a state of excitement, making complex assessment impossible.

Despite these limitations, our results provide meaningful information in interpreting the BIS and entropy in children with cerebral palsy, in whom entropy may help distinguish consciousness from unconsciousness as well as the BIS. Considering that entropy describes the irregularity, complexity, or unpredictability of the EEG [20] and assuming that BIS reflects the EEG characteristics of the general population (see the Appendix A), entropy may be useful for CP patients. Furthermore, looking to exhibiting hysteresis, both BIS and entropy can be said to suggest that the limits of the EEG signal processing for real-time evaluation of consciousness [21].

In conclusion, the absolute state entropy values obtained while awake, as well as the response entropy, state entropy, and BIS at 3% end-tidal sevoflurane concentration are lower in children with cerebral palsy than in those without. The BIS is better correlated than entropy at the point of LOR during sevoflurane induction in children without cerebral palsy, but not in children with cerebral palsy. A different pattern of background brain activity in children with cerebral palsy may influence entropy more than the BIS. Further evaluation of the correlation between brain activity in cerebral palsy and DOA monitoring is worthwhile, utilizing not only EEG but also other modalities.

## Figures and Tables

**Figure 1 entropy-21-00498-f001:**
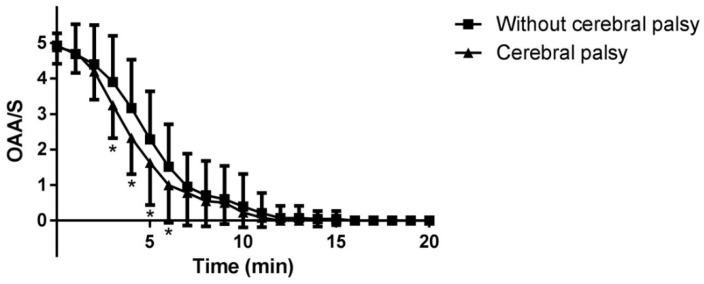
The change of Observer’s Assessment of Alertness/Sedation rating (OAA/S) scale during the induction period. Data are shown as mean ± SD. * *p* < 0.05 compared with without cerebral palsy.

**Figure 2 entropy-21-00498-f002:**
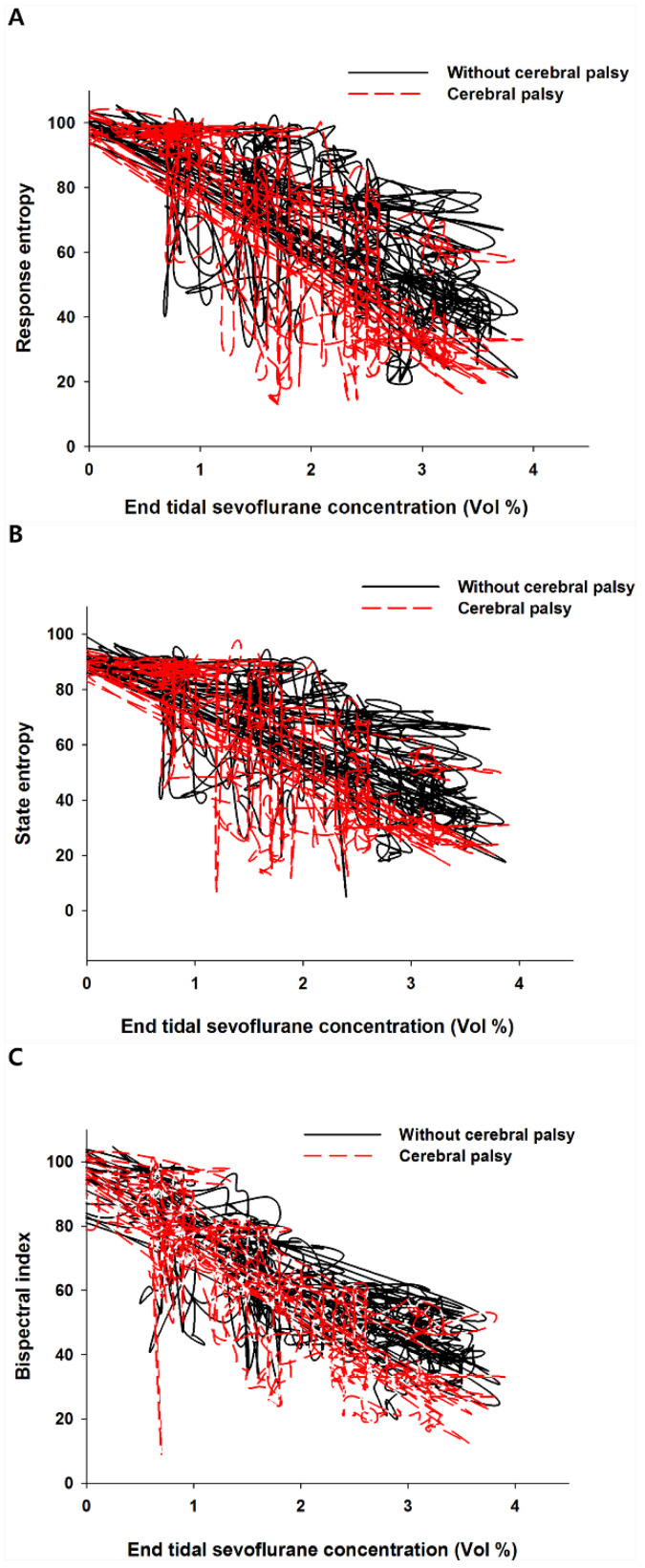
Anesthetic depth and end-tidal sevoflurane concentration during the induction period. This figure shows hysteresis on (**A**) response entropy, (**B**) state entropy, or (**C**) bispectral index with regard to end-tidal sevoflurane concentration. The red dotted lines refer the children with cerebral palsy; the black solid lines refer to the children without cerebral palsy.

**Figure 3 entropy-21-00498-f003:**
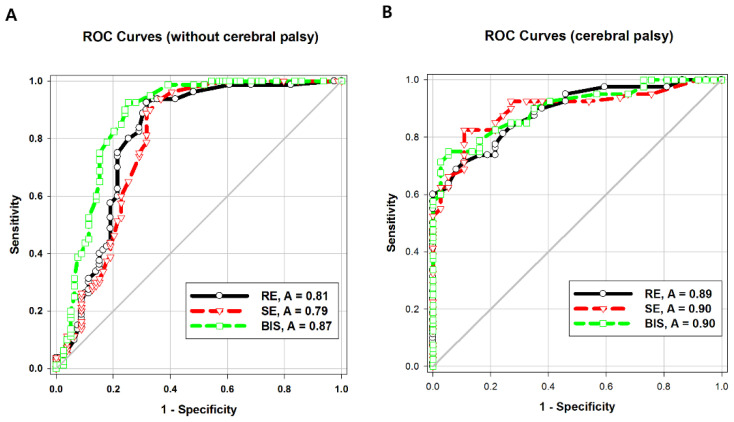
Receiver-operating characteristic (ROC) curves to demonstrate the ability of response entropy (RE), state entropy (SE), and bispectral index (BIS) to predict unresponsiveness in (**A**) children without cerebral palsy or (**B**) children with cerebral palsy.

**Figure 4 entropy-21-00498-f004:**
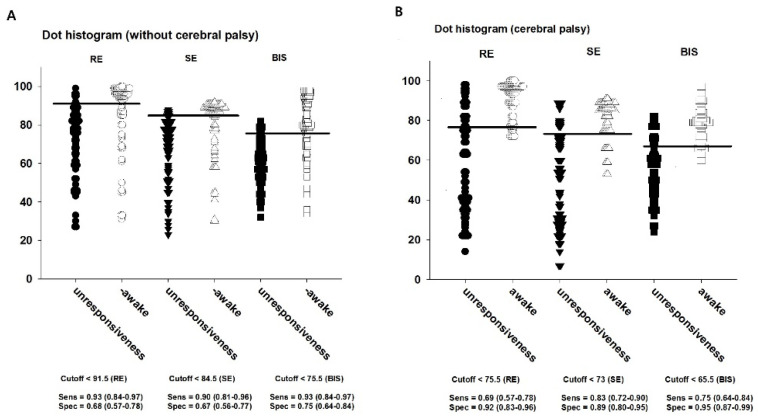
Dot histogram representation of response entropy (RE), state entropy (SE) and bispectral index (BIS) compared with unresponsiveness or awake in the (**A**) children without cerebral palsy or (**B**) children with cerebral palsy. Data are shown as median ± interquartile range. The horizontal lines and the tables to the bottom of the graph represent the cut-off values for RE, SE, and BIS, and the corresponding sensitivity/specificity.

**Table 1 entropy-21-00498-t001:** Demographic data.

	Group C (n = 42)	Group CP (n = 40)
Age (year)	10.2 ± 2.9	11.1 ± 2.0
Sex (M/F)	20/22	14/26
Weight (kg)	25.6 ± 10.8	23.0 ± 9.4
Height (cm)	116.0 ± 18.7	118.6 ± 18.4

Values are mean ± SD or number of patients. Group C: Children without cerebral palsy, Group CP: Children with diplegic cerebral palsy.

**Table 2 entropy-21-00498-t002:** RE, SE, BIS values and end-tidal (ET) sevoflurane concentrations during the induction periods.

		Group C (n = 42)	Group CP (n = 40)
Awake state	RE	97.5 ± 1.7	96.8 ± 2.0
	SE	89.5 ± 2.9	86.9 ± 2.0 *
	BIS	94.2 ± 4.9	93.1 ± 5.7
ET Sevoflurane (vol%) (beginning excitement)		1.6 ± 0.1	1.3 ± 0.1 *
ET Sevoflurane (vol%) (unresponsiveness)		1.7 ± 0.4	1.5 ± 0.3
At 3 vol% of ET Sevoflurane	RE	53.3 ± 11.1	39.1 ± 12.0 *
	SE	47.7 ± 9.9	34.8 ± 10.2 *
	BIS	47.6 ± 6.2	37.6 ± 9.1 *

Values are mean ± SD. Group C: Children without cerebral palsy, Group CP: Children with spastic diplegic cerebral palsy, RE: Response entropy, SE: State entropy, BIS: Bispectral index; ^*^
*p* < 0.05 compared with Group C.

**Table 3 entropy-21-00498-t003:** Area under the curve (AUC) for responsiveness present.

	AUC	
	Group C (n = 42)	Group CP (n = 40)
RE	0.81 ± 0.03	0.89 ± 0.02 *
SE	0.79 ± 0.03	0.90 ± 0.02 *
BIS	0.87 ± 0.03 ^+,‡^	0.90 ± 0.02

Values are mean ± SD. Group C: Children without cerebral palsy, Group CP: Children with spastic diplegic cerebral palsy, RE: Response entropy, SE: State entropy, BIS: Bispectral index; ^*^
*p* < 0.05 compared with Group C, ^+^
*p* < 0.05 compared with RE, ^‡^
*p* < 0.05 compared with SE.

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
