# Peer review of "A Comparison of Bispectral Index and Entropy During Sevoflurane Anesthesia Induction in Children with and without Diplegic Cerebral Palsy"

_entropy, 2019, doi:10.3390/e21050498_

Reviewer 1 Report

My only concern is the description of the groups. In CP group there is lack of information what type of CP represent patients (spastic, atactic), which can influence the results (what clearly the authors describe in their discussion). There is also no information about the patients in the reference groups: what was their health status, and possible handicap for the results.

Author Response

Response to Reviewer 1 Comments 

Point 1: My only concern is the description of the groups. In CP group there is lack of information what type of CP represent patients (spastic, atactic), which can influence the results (what clearly the authors describe in their discussion).

Response 1:

I agree your opinion that the type of CP was important. In fact, although the most type of CP was spastic but there were also other types of CP.

I described the type distribution of CP in the result section. And I also added this to limitations in the discussion part.

Line 116: “The most type of CP was spastic (31 patients; 77.5%). 7 patients (17.5%) had dyskinetic CP and the other 2 patients (5%) had ataxic CP.

Line 229: “First, CP types in the study were not homogeneous. Although the most type was spastic, the other types including dyskinetic and ataxic CP may influence the results and make the interpretation difficult. However, since all the children had not taken anti-epileptic medications, the effect might not be significant.

Point 2: There is also no information about the patients in the reference groups: what was their health status, and possible handicap for the results.

Response 2: The information about the patients in the reference group was added in the result section considering your recommendation.

 Line 117: “In the group C, there were 2 patients with achodroplasia. And the other 40 patients were of ASA I and no medical history.

Thank you very much for your kind review and recommendations. It was very helpful for revising and improving the paper.

Reviewer 2 Report

 This study was to examine the correlation of BIS and entropy (state and response) on the different concentration of sevoflurane in the child with or without diplegic cerebral palsy. This report suggests that BIS appears to be better correlated with loss of consciousness than entropy in the child without cerebral palsy, whereas BIS seems to be not significantly different form entropy in terms of correlation with state of loss of consciousness in the child with diplegic cerebral palsy. Thus, this report suggests that entropy may be helpful  to differentiate consciousness from unconsciousness. However, this study has some limitations such as calculation of the number of patients in each group (power analysis).

Following comments would be addressed:

Line 93 - 103 Please describe how to calculate the number of patient in each group in this prospective study.

Line 95: Please describe the detailed data (example: BIS, RE and SE et al) to be analyzed using each statistical methods (example: T-test and ANOVA). For example: The comparison of BIS in both was performed using Student’s t-test.

Line 96: Please change “and standard deviation” to (standard deviation)

Line 99: Please change Receiver operating characteristic (ROC) to ROC.

Line 116: is there any significant difference of BIS and entropy in the both group at end-tidal sevoflurane concentration (1 and 2vol%) during induction?

Figure 1: Please move asterisk downward below cerebral palsy.

Line 127-129: Please add “ Data are shown as mean ±SD.”

Line 127-129: Please delete “Group C:----with diplegic cerebral palsy” and correct “* P < 0.05 compared with Group C” to * P < 0.05 compared with without cerebral palsy”.

Line 142: Please change Figure 3a,b to Figure 3A and B.

Line 149-150: Please change (a) and (b) to A and B, respectively.

Line 153: Please change Figure 4a,b to Figure 4A and B.

Line 155-158 (Figure 4 legend): Please add “Data are shown as median ± interquartile range”. Please change (a) and (b) to A and B, respectively.

 Line 157-158: where are cut-off values regarding “the tables to the bottom of the graph represent the cut-off values for RE, SE and BIS, and the corresponding sensitivity/specificity”?

Author Response

Response to Reviewer 2 Comments

This study was to examine the correlation of BIS and entropy (state and response) on the different concentration of sevoflurane in the child with or without diplegic cerebral palsy. This report suggests that BIS appears to be better correlated with loss of consciousness than entropy in the child without cerebral palsy, whereas BIS seems to be not significantly different form entropy in terms of correlation with state of loss of consciousness in the child with diplegic cerebral palsy. Thus, this report suggests that entropy may be helpful to differentiate consciousness from unconsciousness. However, this study has some limitations such as calculation of the number of patients in each group (power analysis).

Following comments would be addressed:

Line 93 - 103 Please describe how to calculate the number of patient in each group in this prospective study.

Response 1: Thank you for your good and sharp review.  The contents for sample size calculation were added to the revised manuscript.

 Line 93: A power analysis suggested that a minimum sample size of 40 patients for each group would be required with a significance level of 5% to achieve a power of 80%. It was calculated from SE values at 3% sevoflurane in preliminary data (Effect size d = 0.636 calculated from 38.6 ± 14.3 of SE in CP versus 46.3 ± 9.4 of SE in non-CP patients). To allow for an exclusion rate, the study population was prospectively set at 90 patients.

 Line 95: Please describe the detailed data (example: BIS, RE and SE et al) to be analyzed using each statistical methods (example: T-test and ANOVA). For example: The comparison of BIS in both was performed using Student’s t-test.

 Response 2: Thank you for the constructive comment. The manuscript was revised as below described.

Line 100: The analysed data were tested for normality using the Kolmogorov-Smirnov test. Either a parametric or non-parametric analysis was performed depending upon the results of the Kolmogorov-Smirnov analysis. Data expressed as mean ± SD were compared using independent t-tests or Mann-Whitney U tests. And data expressed as the number of patients were compared using chi-square analysis or Fisher’s exact test where appropriates.

Line 96: Please change “and standard deviation” to (standard deviation)

Line 99: Please change Receiver operating characteristic (ROC) to ROC.

Response 3: The manuscript was revised as your recommendation.

Line 116: is there any significant difference of BIS and entropy in the both group at end-tidal sevoflurane concentration (1 and 2vol%) during induction?

Response 4: There was no significant difference in BIS or entropy between the two group at 1 or 2 vol% sevoflurane. And we did not find any difference in state entropy and BIS in the same period.

Figure 1: Please move asterisk downward below cerebral palsy.

Line 127-129: Please add “ Data are shown as mean ±SD.”

Line 127-129: Please delete “Group C:----with diplegic cerebral palsy” and correct “* P < 0.05 compared with Group C” to * P < 0.05 compared with without cerebral palsy”.

Response 5: Thank you for your considerate review. The figure 1 and its legend were revised as your recommendation.

Line 142: Please change Figure 3a,b to Figure 3A and B.

Line 149-150: Please change (a) and (b) to A and B, respectively.

Line 153: Please change Figure 4a,b to Figure 4A and B.

Line 155-158 (Figure 4 legend): Please add “Data are shown as median ± interquartile range”. Please change (a) and (b) to A and B, respectively.

Response 6: As your opinion, the manuscript and figure legend were revised.

 Line 157-158: where are cut-off values regarding “the tables to the bottom of the graph represent the cut-off values for RE, SE and BIS, and the corresponding sensitivity/specificity”?

Response 7: Thank you for your precise comment. I revised the figure 4 to represent the cut-off values for RE, SE and BIS, and the corresponding sensitivity/specificity.

Thank you very much for your kind review and recommendations. It was very helpful for revising and improving the paper.

Round  2

Reviewer 2 Report

This revised manuscript was improved and is acceptable.

Author fully responded to the reviewer’s comments.

Author Response

I heartily thank you for your careful review.

For your guidance, the manuscript was improved.